# Investigating the Causal Effects of Exercise-Induced Genes on Sarcopenia

**DOI:** 10.3390/ijms251910773

**Published:** 2024-10-07

**Authors:** Li Wang, Song Zhang

**Affiliations:** 1Institute of Sports Medicine and Health, Chengdu Sport University, Chengdu 610041, China; 2College of Animal Science and Technology, Northwest A&F University, Yangling 712100, China; zhangsong@nwafu.edu.cn

**Keywords:** exercise, skeletal muscle aging, sarcopenia, transcriptome, two-sample Mendelian randomization

## Abstract

Exercise is increasingly recognized as an effective strategy to counteract skeletal muscle aging and conditions such as sarcopenia. However, the specific exercise-induced genes responsible for these protective effects remain unclear. To address this, we conducted an eight-week aerobic exercise regimen on late-middle-aged mice and developed an integrated approach that combines mouse exercise-induced genes with human GWAS datasets to identify causal genes for sarcopenia. This approach led to significant improvements in the skeletal muscle phenotype of the mice and the identification of exercise-induced genes and miRNAs. By constructing a miRNA regulatory network enriched with transcription factors and GWAS signals related to muscle function and traits, we focused on 896 exercise-induced genes. Using human skeletal muscle *cis*-eQTLs as instrumental variables, 250 of these exercise-induced genes underwent two-sample Mendelian randomization analysis, identifying 40, 68, and 62 causal genes associated with sarcopenia and its clinical indicators—appendicular lean mass (ALM) and hand grip strength (HGS), respectively. Sensitivity analyses and cross-phenotype validation confirmed the robustness of our findings. Consistently across the three outcomes, *RXRA*, *MDM1*, *RBL2*, *KCNJ2*, and *ADHFE1* were identified as risk factors, while *NMB*, *TECPR2*, *MGAT3*, *ECHDC2*, and *GINM1* were identified as protective factors, all with potential as biomarkers for sarcopenia progression. Biological activity and disease association analyses suggested that exercise exerts its anti-sarcopenia effects primarily through the regulation of fatty acid oxidation. Based on available drug–gene interaction data, 21 of the causal genes are druggable, offering potential therapeutic targets. Our findings highlight key genes and molecular pathways potentially responsible for the anti-sarcopenia benefits of exercise, offering insights into future therapeutic strategies that could mimic the safe and mild protective effects of exercise on age-related skeletal muscle degeneration.

## 1. Introduction

Skeletal muscle aging refers to the natural physiological process in which the structure and function of skeletal muscles gradually deteriorate with age. Sarcopenia is a common pathological manifestation of this process, frequently observed in the elderly, leading to reduced functional capacity and diminished quality of life in older adults [1,2]. Characterized by a significant reduction in skeletal muscle mass and strength [3,4,5], sarcopenia is typically diagnosed and assessed using appendicular lean mass (ALM) and hand grip strength (HGS) metrics [6,7]. Currently, there are no specific medications to treat sarcopenia effectively. Hormonal therapies, such as testosterone and selective androgen receptor modulators, are the primary treatments, but they come with significant side effects [8]. Therefore, there is an urgent need to understand the molecular and cellular mechanisms of sarcopenia to identify safe therapeutic targets and develop interventions for skeletal muscle aging and sarcopenia.

Exercise is the primary non-pharmacological intervention to combat skeletal muscle aging and sarcopenia, significantly improving muscle function, including muscle mass and strength [9,10]. These improvements are not limited to physical attributes but also profoundly affect the molecular mechanisms within muscle cells [11,12,13]. Despite the well-known benefits of exercise, the specific molecular mechanisms behind its anti-aging effects on skeletal muscle remain unclear. Studies have shown that skeletal muscle miRNAs are highly responsive to exercise stimuli [14,15,16], suggesting that the target genes of these miRNAs may play crucial roles in the effects of exercise. Exploring these intermediary genes will help understand how exercise mitigates skeletal muscle aging and related diseases and may provide potential therapeutic targets. Drugs targeting these genes could become effective exercise mimetics, simulating the physiological and biochemical changes naturally induced by exercise, thereby aiding in the prevention and treatment of sarcopenia with better safety profiles.

With the increase in population genetic data, Mendelian randomization (MR) has become a powerful method for identifying potential therapeutic targets or biomarkers for diseases or traits [17,18]. MR uses genetic variation as an instrumental variable, similar to the random allocation mechanism in randomized controlled trials, to estimate the causal relationship between exposure (e.g., gene expression, protein levels, or metabolites) and outcomes (e.g., diseases and traits). This approach effectively reduces confounding bias and reverse causation, providing more reliable causal inferences [19,20,21]. Two-sample MR uses data from different studies separately as exposure and outcome, enhancing each study’s flexibility and efficiency [22]. By using different sets of instrumental variables, two-sample MR can evaluate the impact of the same exposure on different outcomes or the effects of different exposures on the same outcome, offering a more comprehensive causal relationship map.

Recently, two-sample MR analyses have been used to systematically identify potential drug targets for sarcopenia, demonstrating the method’s potential in accelerating drug development for sarcopenia [23,24,25]. However, the exposure data used by Chen et al. [23] and Jiang et al. [25] were derived from blood rather than skeletal muscle pQTL data, leading to a lack of specificity and relevance in the causal relationships between exposure and outcomes. Although Yin et al. integrated two types of exposure data (two blood pQTL datasets and one skeletal muscle eQTL dataset) [24], cross-validation among these different tissue-derived data remains questionable. Their study was limited to existing druggable genes, and the GTEx eQTL data used were not from the latest v8 version. More critically, these studies focused on broad screening for potential drug targets for sarcopenia, with low interpretability of the prevention mechanisms, which may limit subsequent basic and clinical research.

In this context, we leveraged the prior knowledge that exercise can effectively improve skeletal muscle aging and sarcopenia. We analyzed the causal effects of exercise-induced genes on traits related to sarcopenia and explored the mechanisms of these genes, as well as their potential as therapeutic targets of sarcopenia. To demonstrate the anti-aging effects of exercise, we specifically designed an aerobic exercise intervention in mice, ranging from late middle age (14 months) to old age (16 months), as previously described for aged mice [26]. We developed an integrated approach using two-sample MR to combine mouse exercise-induced genes from the miRNA regulatory network with human GWAS data, evaluating the causal effects of the expression of these genes on three sarcopenia-related traits: sarcopenia, ALM, and HGS. To maintain interpretability and avoid potential pleiotropy, we used only skeletal muscle-derived GTEx *cis*-eQTLs [27] as instrumental variables (IVs).

## 2. Results

### 2.1. Exercise-Induced Histological and Transcriptome Differences in Aging Skeletal Muscle

To investigate the anti-aging effects of exercise on skeletal muscles, we implemented an exercise intervention in mice transitioning from late middle age (14 months) to old age (16 months), using sedentary mice as controls. Hematoxylin and eosin (H&E) staining was performed for histological analysis to assess the impact of exercise. In the exercise group (oldEx), we observed a notable increase in muscle fiber thickness and a reduction in intermuscular fat, evidenced by decreased spaces between muscle fibers (Figure 1a). This finding aligns with the well-established concept that exercise promotes muscle growth (muscle protein synthesis) and increases energy expenditure (oxidative metabolism of intermuscular fat) [28,29], highlighting the effectiveness of the intervention. Additionally, RNA-Seq analysis demonstrated high within-group reproducibility (Figure 1b). A total of 1297 differentially expressed genes (DEGs) were identified (adjusted *p*-value < 0.05, Figure 1c, Appendix A). Among these, genes encoding myosins, the primary contractile proteins responsible for converting chemical energy from ATP hydrolysis into mechanical energy, such as *Myh1*, *Myh2*, and *Myh3* [30,31], were upregulated. Furthermore, marker genes associated with fibro/adipogenic progenitors (FAPs), the cells from which intermuscular fat originates, including *Fap*, *Bmper*, and *Dpep1* [26], as well as markers for mature adipocytes like *Fasn* [26], were downregulated. These DEGs are significantly enriched in biological processes related to muscle development (muscle system processes and muscle cell differentiation) and cellular signaling pathways involving muscle function (motor proteins and focal adhesion). They are also enriched in biological processes (fatty acid metabolic process and lipid catabolic process) and cellular signaling pathways (PPAR signaling pathway and fatty acid degradation) associated with lipid metabolism (Figure 1d,e). These findings suggest a potential causal link between exercise-induced tissue alterations and transcriptomic changes, emphasizing the need to further explore their underlying molecular mechanisms.

### 2.2. Exploring the Potential miRNA–Target Network Response to Exercise

Given the widespread response of miRNA regulatory pathways to exercise stimuli, we aimed to construct a miRNA regulatory network mediating the anti-aging effects of exercise on skeletal muscle. Accordingly, we performed miRNA-Seq on the skeletal muscle samples mentioned earlier. All samples had an average sequencing depth of nearly 10 million reads, with insert sizes symmetrically distributed around 22 nucleotides, consistent with previously reported reference values [32,33]. In our study, we identified 44 differentially expressed miRNAs (DEmiRs), as shown in Figure 2a and detailed in Appendix A. This set includes 27 upregulated and 17 downregulated miRNAs. Among these are several muscle-specific miRNAs (also known as myomiRs) that have a broad impact on muscle biological processes, including miR-206-3p, miR-486a-5p/486b-5p/486a-3p, and miR-499-5p [34,35,36,37,38], as well as others with significant regulatory roles in muscle functions, such as miR-23a-3p/23b-3p and miR-155-5p [39,40,41,42,43]. Subsequently, we constructed a targeted regulatory network between the DEGs and DEmiRs based on inverse expression changes and target degradation predictions (as forecasted by the starBase database) [44]. The final regulatory network comprises 33 miRNAs, 896 predicted target genes, and 3771 regulatory relationships (Figure 2b, Appendix A).

We identified the network’s upstream transcription factors (TFs) using the TransmiR database [45]. The top five TFs with the highest enrichment significance (hypergeometric test, *p*-value < 0.05, Figure 2c) are closely associated with various skeletal muscle biological processes. These include SREBF2, which regulates intramuscular fat balance [46]; RBPJ, maintaining muscle progenitor cells [47]; STAT5A, conveying growth hormone signals in muscles [48]; BCL6, controlling myoblast proliferation, differentiation, and apoptosis [49,50]; and EBF2, regulating muscle relaxation [51]. These results indirectly support the functional specificity of this miRNA regulatory network in skeletal muscle. Additionally, it is widely recognized that a disproportionate clustering of trait-associated SNPs in genomic regions near specific gene sets, rather than a random distribution, indicates a potential genetic linkage between the trait and the gene set [52,53,54]. To evaluate the genetic association between our network and aged muscle disorders, we compiled SNPs linked with human skeletal muscle aging, muscular diseases, and other complex traits from the GWAS catalog database [55]. All these SNP–trait associations had genome-wide significance (*p*-value ≤ 5 × 10^−8^), encompassing 76,678 human SNPs linked to 47 traits (Appendix A). Following previous studies [53,56,57], we assessed the enrichment of these SNPs within the transcriptional regulatory regions (TRRs, ±20 kb around transcription start sites) of the network members by mapping these SNPs to the orthologous sequences of the TRRs in the human genome. We observed that SNPs associated with waist–hip ratio, monocyte count, appendicular lean mass, heart disease, and vascular disease were significantly enriched in the TRRs of this network (hypergeometric test, FDR < 0.05, Figure 2d). These traits are related to metabolic health [58,59], inflammatory response [60,61], muscle mass, and cardiovascular disease [62], all of which are closely linked to skeletal muscle aging. Overall, we believe this network may respond to exercise stimuli and mediate the effects against skeletal muscle aging.

### 2.3. Calculating the Causal Effects of Exercise-Induced Genes on Sarcopenia Outcomes

Based on the above results, we hypothesize that the miRNA regulatory network mediates the anti-aging effects of exercise on skeletal muscle. To investigate the causal relationship between this network’s target genes and human sarcopenia-related traits, we leveraged skeletal muscle eQTL summary statistics from the Genotype-Tissue Expression (GTEx) project [27], GWAS summary statistics for sarcopenia [63] and ALM [64] from the GWAS catalog, and HGS data [65] from the OpenGWAS database. Using a two-sample MR approach, we estimated the causal effect sizes of these genes on these outcomes (Figure 3a). To avoid potential pleiotropy, we used only *cis*-eQTLs located within ±1 Mb of the TSSs as IVs. After rigorous screening (Figure 3a), 248 genes for sarcopenia, 250 genes for ALM, and 249 genes for HGS were retained for testing. Using the Wald ratio (for a single IV) or inverse variance weighted (IVW, for two or more IVs) methods, we identified 40 causal genes for sarcopenia (Bonferroni-corrected *p* ≤ 2.02 × 10^−4^ = 0.05/248 genes; Figure 3b), 68 for ALM (Bonferroni-corrected *p* ≤ 2.00 × 10^−4^ = 0.05/250 genes; Figure 3c), and 62 for HGS (Bonferroni-corrected *p* ≤ 2.00 × 10^−4^ = 0.05/249 genes; Figure 3d), with consistency across four other methods (MR–Egger, weighted median, simple mode, and weighted mode), as shown in Appendix A. Further sensitivity analyses confirmed the robustness of the primary MR findings. Our IVs did not exhibit horizontal pleiotropy (MR–Egger intercept, *p* > 0.05) or heterogeneity (Cochran Q statistics, *p* > 0.05), as shown in Appendix A. Additionally, our data passed the Steiger filter test (*p* < 0.05), supporting our hypothesis that the expression of these genes leads to the outcome of sarcopenia, thereby ruling out the possibility of reverse causation.

From a causal relationship perspective, 21 genes were positively and 19 genes were negatively associated with the risk of sarcopenia; 32 genes were positively and 36 genes were negatively associated with ALM; and 29 genes were positively and 33 genes were negatively associated with HGS. Notably, four ALM-related causal genes (*RXRA*, *COL15A1*, *AURKA*, and *SMAD3*) identified in our study have been previously reported [24] with consistent causal relationships (whether positive or negative), as documented in earlier research. Among the 113 non-redundant causal genes of the three sarcopenia-related outcomes, 37 were causal for two outcomes, and 10 were causal for all three outcomes (Figure 3e). Given that reductions in ALM and HGS are common features of sarcopenia [6,7], these 10 shared genes exhibited consistent causal effects: 5 genes (*RXRA*, *MDM1*, *RBL2*, *KCNJ2*, *ADHFE1*) were positive for sarcopenia and negative for ALM and HGS, while 5 genes (*NMB*, *TECPR2*, *MGAT3*, *ECHDC2*, *GINM1*) were negative for sarcopenia and positive for ALM and HGS (Figure 4a–c). This underscores the reliability of our MR analysis and their potential as biomarkers for sarcopenia.

### 2.4. Unveiling Biological Mechanisms and Drug Interactions of These Causal Genes

Based on previous findings, we propose that exercise provides anti-sarcopenia benefits by regulating the above 113 genes linked to the condition. Investigating the biological mechanisms and drug interactions of these genes could lead to therapies that mimic the effects of exercise, benefiting elderly individuals with limited mobility. Using the Enrichr platform [66], we analyzed the biological roles of these genes through GO Biological Process and WikiPathway tools, revealing a primary involvement in fatty acid oxidation (Figure 5a,b). ClinVar analysis on Enrichr also showed a significant association between these genes and disorders related to fatty acid metabolism (Figure 5c). These findings suggest that the anti-sarcopenia effects of exercise may be mediated through the regulation of fatty acid oxidation pathways. Further analysis using the DSigDB [67] option on Enrichr identified four drugs (Aflatoxin B1, Cyclosporin A, Menadione, and Acetaminophen) that significantly alter the expression of these genes (Figure 5d), highlighting their potential as therapeutic targets for sarcopenia. However, since these drug–gene interactions are based on drug-induced gene expression changes, and most of these drugs have significant side effects [68,69,70,71], their potential for developing sarcopenia treatments is limited. To improve the likelihood of identifying viable drug targets for sarcopenia, we filtered the causal genes to identify druggable genes—those whose protein products have drug-binding sites. We downloaded 2531 druggable genes compiled by Yin et al. [24], which represent an intersection of data from the DGIdb database [72] and the study by Finan et al. [73], providing a reliable source. Among the identified causal genes, we found that 21 are classified as druggable (Figure 5e), with nearly 800 drugs targeting these genes (Appendix A). These 21 causal genes are considered potential therapeutic targets for sarcopenia. Further validation of drugs targeting these genes could aid in developing therapies that mimic the anti-sarcopenia effects of exercise.

## 3. Discussion

To ensure robust anti-aging effects, we specifically implemented exercise intervention on mice during the transition from late middle age to old age. Previous research has shown that the adaptive changes in the skeletal muscle transcriptome to exercise tend to diminish with advancing age [74,75]. This phenomenon suggests that exercise interventions may not consistently yield significant transcriptomic changes in elderly individuals. In fact, a comprehensive study has demonstrated that moderate-intensity aerobic training can reduce intermuscular fat content in individuals aged 18 to 65 [76]. However, in frail older populations, exercise may inadvertently lead to adverse changes in intermuscular fat content [76]. This variability may stem from reduced compliance and increased physiological stress due to multiple comorbidities in frail elderly individuals [77], confounding the clear effects of exercise interventions. Therefore, we propose that implementing aerobic exercise interventions in middle-aged individuals on the verge of aging provides a more favorable approach for elucidating the components through which exercise exerts its anti-aging effects. In line with this rationale, we subjected 14-month-old mice to an 8-week aerobic exercise training regimen, considering that 16 months is the threshold for entering the old-age stage in mice [26].

To elucidate the molecular pathways of this exercise intervention, we analyzed exercise-induced transcriptome perturbations. These DEGs are involved in skeletal muscle development and fatty acid oxidation, consistent with observed phenotypic improvements such as increased muscle fiber size and reduced intramuscular fat. Additionally, given the extensive and profound influence of miRNAs, such as myomiRs, in skeletal muscle and their rapid response to exercise stimuli, we performed miRNA-seq and constructed an exercise-induced miRNA regulatory network that further refined the initial DEGs. Based on the inclusion of well-known miRNAs, coding genes, and enriched upstream TFs, this network appears to be involved in diverse skeletal muscle biological processes, demonstrating a strong skeletal muscle specificity. Indeed, by further integrating summary statistics from GWAS data, we found that this network is significantly enriched in SNPs associated with multiple muscle phenotypes and age-related muscle disorder indicators, supporting its close association with skeletal muscle aging at the population genetics level. Based on these findings, we hypothesize that the DEGs within this network may mediate the anti-aging benefits of exercise on skeletal muscle.

We have identified a new set of MR causal genes for sarcopenia that may mediate the anti-sarcopenia effects of exercise. The robustness of our MR results was ensured through primary MR analysis, Bonferroni correction, heterogeneity analysis, pleiotropy analysis, and Steiger filtering. Ultimately, 250 unique genes were tested, and 113 causal genes related to sarcopenia traits were identified. The high positive detection rate (113/250) compared to another recent study [24] could be attributed to several factors: the use of exercise-induced genes in aging skeletal muscle as the basis for evaluation, the strong association between skeletal muscle IVs and sarcopenia traits, the use of the latest eQTL data, and the adoption of a relatively lenient LD (linkage disequilibrium) filtering threshold. The first two factors ensured a high relevance between exposure genes and sarcopenia outcomes, while the latter two allowed for the inclusion of more eligible IVs. The causal effects of these 113 genes were calculated using an average of 14 IVs per gene, compared to typically just 1 IV per gene in the other recent study [24]. The use of more IVs provides a more comprehensive reflection of the relationship between exposure and outcome, reduces the risk of bias, and enhances statistical power [20].

Our research has identified new potential biomarkers for sarcopenia. Sarcopenia, ALM, and HGS are interrelated phenotypes. The distribution of these 113 causal genes across the three outcomes reflects their specificity and commonality. Among them, 66 causal genes were detected in only one of the three outcomes, suggesting that certain genes may primarily influence muscle volume, while others may have a greater impact on muscle strength. Meanwhile, 47 causal genes were found in at least two outcomes, indicating their role in shared biological pathways across these traits. Notably, 10 genes exhibited consistent causal effects across all three outcomes, potentially playing key roles in core biological processes related to muscle mass and strength through pathways such as metabolic regulation (*RXRA* [78], *ADHFE1* [79], *MGAT3* [80], *ECHDC2* [81], *NMB* [81]), cell cycle regulation (*MDM1* [82], *RBL2* [83]), signal transduction (*KCNJ2* [84], *GINM1* [84]), and autophagy/cell clearance (*TECPR2* [84]). Monitoring changes in these genes could provide valuable insights for the early detection and diagnosis of sarcopenia, particularly by identifying high-risk individuals through gene expression or variation analysis before clinical symptoms become apparent.

Our study sheds light on the primary molecular pathways by which exercise confers protective effects against sarcopenia and offers potential therapeutic targets. To gain a comprehensive understanding of the exercise-induced anti-sarcopenia mechanisms, we utilized the Enrichr platform to analyze the biological processes, cellular pathways, and clinical diseases associated with all 113 causal genes. We found that these genes are primarily involved in fatty acid oxidation activities and are closely linked to the development of fatty acid metabolic disorders. Existing research demonstrates that exercise-induced fatty acid oxidation produces a range of beneficial physiological effects that enhance skeletal muscle health on multiple levels. These effects include boosting energy metabolism to maintain skeletal muscle function and mass [85], reducing lipid accumulation to protect skeletal muscle from chronic inflammation [86], and improving mitochondrial function to safeguard skeletal muscle from oxidative damage [87]. Our findings suggest that fatty acid oxidation is a key pathway through which exercise exerts its anti-sarcopenia effects and highlight 113 potential intermediary genes that mediate this pathway. Based on the available drug–gene interaction data, we identified 21 causal genes that are druggable, making them potential therapeutic targets for sarcopenia. The DGIdb database records nearly 800 drugs that target these genes, providing a foundation for developing new treatments or repurposing existing drugs (i.e., drug repositioning) to combat sarcopenia. Given the role of these druggable genes in mediating exercise effects, targeting them could lead to the development of safe and mild therapeutic strategies for sarcopenia.

The strategy of integrating mouse and human data in this study is generally appropriate, although it does have some limitations and challenges. Mice are the most commonly used model organisms in human disease research and have been extensively utilized to understand the mechanisms by which exercise combats skeletal muscle aging [88,89]. Mice share a high degree of genetic homology with humans, and their organs and systems function similarly, allowing them to exhibit various diseases analogous to those in humans [90]. Obtaining preliminary data from mouse experiments and then combining them with GWAS summary statistics of human diseases or traits is an effective approach to identify key factors. For example, Li et al. integrated mouse liver co-expression networks with human lipid GWAS data to identify causal genes related to cholesterol and lipid metabolism [91]. Similarly, our study identified causal genes for sarcopenia by integrating exercise-induced genes from aging mouse skeletal muscle with human GWAS data related to sarcopenia. The results indicate a strong alignment between the data from the two species. For instance, the exercise-induced genes identified in aging mouse skeletal muscle are significantly enriched in human-derived GO and KEGG annotations related to muscle tissue development and fatty acid oxidation (Figure 1d,e). Additionally, the exercise-induced miRNA regulatory network is closely linked to diseases or traits associated with human skeletal muscle aging (Figure 2d). However, there are some limitations to consider. Cross-species differences mean that the responses to exercise in mice and humans may not be entirely consistent, indicating potential differences in the exercise-induced genes and regulatory networks between the two species. Furthermore, the shorter lifespan of mice and the highly controlled experimental conditions of short-term interventions do not adequately simulate the long-term effects of exercise in the complex environment of human aging. The contribution of this study lies in its focus on specific research targets, providing ten potential biomarkers for sarcopenia and twenty-one potential therapeutic targets. Future research could explore the exercise response of these key genes in human skeletal muscle aging and their applications in diagnosing and preventing sarcopenia.

## 4. Materials and Methods

### 4.1. Animal Care and Use

Male wild-type (C57BL/6J) mice at 14 months of age were procured from Chengdu Dossy Experimental Animals Co., Ltd. (Chengdu, China), and housed in standard cages at the Laboratory Animal Center of Chengdu Sport University (Chengdu, Sichuan, China). After an adaptation period of one week, the mice were randomly divided into two groups, each consisting of four mice: the sedentary control group and the aerobic exercise-trained group. The exercise-group mice were subjected to treadmill exercise for 40 min per day at a gradually increasing speed (from 8 m/min to 16 m/min), five days a week, for a duration of two months. Mice in the sedentary group were allowed to eat freely and were not trained. All animal care and experimental procedures adhered to the Guide for the Care and Use of Laboratory Animals, and the research received approval from the Animal Ethics Committee of Chengdu Sport University.

### 4.2. Histological Investigation

The mice were humanely euthanized via spinal dislocation, and subsequently, the quadriceps muscles from their hindlimbs were meticulously dissected. The muscle specimens were then immersed in 4% formalin and later embedded in paraffin. Employing a microtome (Microm HM 550, ThermoFisher Scientific, Waltham, MA, USA), the paraffin sections were skillfully sliced to a thickness of 4 μm. These sections were subjected to H&E staining for histological examination, aimed at assessing the changes in muscle fibers and intermuscular fat. The stained tissue slices were meticulously scanned utilizing a digital slide scanner (Pannoramic SCAN II, 3DHISTECH, Budapest, Hungary).

### 4.3. Sequencing Library Construction

We collected the quadriceps muscles from the old group and the oldEx group of mice for transcriptome sequencing. Total RNA was extracted from these samples using a TRIzolTM reagent (Invitrogen, Carlsbad, CA, USA), followed by rRNA depletion and DNaseI treatment (Qiagen, Hilden, Germany). Both the RNA-seq libraries and miRNA-seq libraries were constructed with the assistance of Novogene Corporation (Beijing, China). The RNA-seq libraries and miRNA-seq libraries were sequenced on the NovaSeq 6000 platform (Illumina, San Diego, CA, USA), generating 150 bp paired-end and 50 bp single-end reads, respectively, for subsequent analysis. All raw sequencing data have been deposited in the Genome Sequence Archive (GSA) under accession number CRA012450.

### 4.4. RNA-Seq Analysis

Raw reads were processed and filtered using Trim Galore and then aligned to the reference mouse genome (fasta, Ensembl release 112) utilizing Hisat2 [92] (v2.2.1). FeatureCounts [93] (v2.0.1) was employed to quantify the reads mapped to each gene based on gene exon coordinates from the gene annotation file (gtf, Ensembl release 112). Gene expression levels were normalized using transcripts per million (TPM) values and further adjusted for library size using edgeR [94] (v3.40.0), resulting in a comprehensive gene expression matrix for all samples. DEGs between groups were identified using DESeq2 [95] (v3.16). The ClusterProfiler R package [96,97] (v4.12.6) was employed to perform GO BP and KEGG signaling pathway enrichment analyses for the DEGs, elucidating their functional roles.

### 4.5. miRNA-Seq Analysis

Raw miRNA-seq reads underwent trimming using Trim Galore with specific parameters: --small_rna --length 18 --max_length 30 --stringency 3. Trimmed reads were aligned to the reference mouse genome using the mapper.pl tool in miRDeep2 [98] (v0.1.3). The expression levels of known miRNAs were estimated using the quantifier.pl tool in miRDeep2. The expression levels of genes were normalized using TPM values and further adjusted for library size using edgeR. This generated a final gene expression matrix for all samples. DEmiRs between groups were identified using DESeq2. The target genes of DEmiRs were predicted using the starBase database (https://rnasysu.com/encori/, accessed on 2 October 2024) [44], supported by Ago CLIP-Seq data.

### 4.6. Enrichment Analysis of TFs and GWAS Traits

To explore the primary upstream TFs in the miRNA regulatory network, data on mouse TF-miRNA regulatory relationships were obtained from the TransmiR v2.0 database [45]. The upstream TFs of the miRNAs in the network were extracted and analyzed. A hypergeometric distribution test (*p*-value < 0.05) was used to evaluate the significant enrichment of upstream TFs for these miRNAs. Additionally, to investigate traits closely associated with this network, we collected trait-related SNPs from the GWAS Catalog database [55]. SNPs located within the network’s TRRs were extracted and analyzed. A hypergeometric distribution test (*p*-value < 0.05) was applied to assess the significant enrichment of GWAS traits among the trait-related SNPs within the network’s TRRs.

### 4.7. Exposure Data and Outcome Data

Skeletal muscle *cis*-eQTL summary data from 588 European individuals were obtained from the GTEx project [27]. GWAS summary data for sarcopenia (GCST90007526) [63] and ALM (GCST90000025) [64] were retrieved from the GWAS Catalog database, while data for HGS (ukb-b-10215) [65] were sourced from the OpenGWAS database. The sarcopenia data come from a GWAS involving 256,523 Europeans (48,596 cases and 207,927 controls) aged 60 and older [63]. The summary statistics were based on the low hand grip strength criteria defined by the European Working Group on Sarcopenia in Older People (EWGSOP) [6], with grip strength thresholds set at <30 kg for males and <20 kg for females. The ALM data were derived from a GWAS analysis of 450,243 Europeans [64], and the HGS data from a GWAS analysis of 461,089 Europeans [65].

### 4.8. Mendelian Randomization Analysis

We conducted a two-sample MR analysis using the TwoSampleMR R package [99], following established protocols from the previous literature [25,100]. In brief, we initiated the analysis by selecting SNPs that met specific criteria to serve as IVs: *cis*-eQTLs (within TSS ± 1 Mb), genome-wide significance level (*p* < 5 × 10^−8^) with the exposure of interest, LD clumping (r^2^ < 0.05 and distance > 10 kb) using the 1000 Genomes reference panel [101], and F-statistic < 10. Ambiguous SNPs were harmonized and excluded from further analysis. Subsequently, we employed the Wald ratio method to estimate effects when a gene was instrumented by a single SNP, and for genes instrumented by two or more SNPs, we utilized the fixed-effect IVW method. To assess the robustness of the associations, we applied additional methods including simple mode, weighted mode, weighted median, and MR–Egger. The MR–Egger intercept test was employed to identify potential unbalanced pleiotropy (horizontal pleiotropy) of IVs. Cochran Q statistics were calculated using both MR–Egger and IVW methods to assess the heterogeneity of IVs. The IVs for these causal genes are listed in Appendix A.

### 4.9. Biological Mechanisms and Druggable Analysis

To gain a comprehensive understanding of the molecular mechanisms underlying the anti-sarcopenia effects of exercise and to identify potential therapeutic drugs that mimic these effects, we submitted all 113 identified sarcopenia causal genes to the Enrichr online platform (https://maayanlab.cloud/Enrichr/, accessed on 2 October 2024) [66]. Using the GO Biological Process and WikiPathway tools, we explored the biological processes and cellular signaling pathways involving these genes. The ClinVar option was employed to reveal the human diseases closely associated with these genes, and the DSigDB option was used to identify drugs that significantly alter their expression. A statistical threshold of padj < 0.05 was applied. The druggable gene set used in this study was sourced from the DGIdb database [72] and research by Finan et al. [73]. DGIdb compiles drug–gene interaction data from various sources, while Finan et al. link GWAS loci related to complex diseases with potential drug targets, aiding in the identification and validation of therapeutic targets. By integrating these two druggable gene sources, we aimed to reveal the druggability of the causal genes identified in our study.

## 5. Conclusions

First, we subjected aging mice to 2 months of aerobic exercise. Second, we identified 1297 exercise-induced DEGs and 44 DEmiRs. Third, we constructed an exercise-induced miRNA regulatory network closely linked to age-related skeletal muscle degeneration. Fourth, from the exercise-induced genes in this network, we identified 113 causal genes associated with sarcopenia-related traits, including 10 potential biomarkers for sarcopenia. Fifth, through an in-depth analysis of these causal genes, we found that exercise likely exerts its anti-sarcopenia effects primarily by regulating fatty acid oxidation pathways. Sixth, we identified twenty-one druggable genes within these causal genes. In summary, based on prior knowledge of the health benefits of exercise on skeletal muscle aging and sarcopenia, we explored the exercise-induced genes and biological pathways that might mediate these protective effects, offering new options for developing safe and gentle exercise-mimicking drugs targeting sarcopenia.

## Figures and Tables

**Figure 1 ijms-25-10773-f001:**
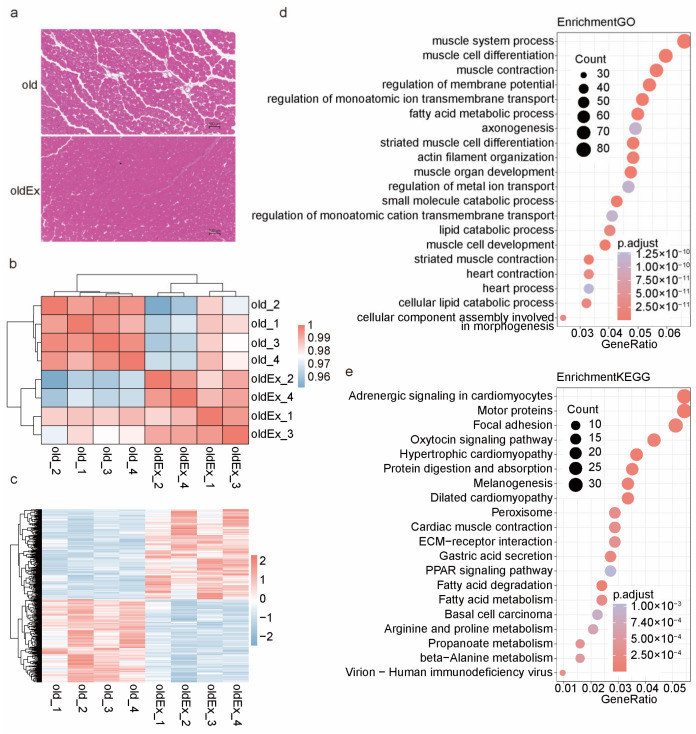
Histological assessment and RNA-Seq analysis of aging skeletal muscle under exercise intervention. (**a**) Representative images of hematoxylin and eosin (H&E)-stained cross-sections of quadriceps muscle from old and oldEx groups. Scale bars, 100 μm. (**b**) Sample similarity clustering. Pairwise correlation (Pearson) among all samples was calculated from the gene expression matrix. (**c**) Heatmap displaying expression profiles of the DEGs (oldEx vs. old, adjusted *p*-value < 0.05) through hierarchical clustering. Each column represents a sample, and each row represents a gene. The color scale indicates the raw Z-score, ranging from red (high expression) to blue (low expression). (**d**,**e**) The top 20 GO biological processes (**d**) and top 20 KEGGs signaling pathways (**e**) enriched by the DEGs.

**Figure 2 ijms-25-10773-f002:**
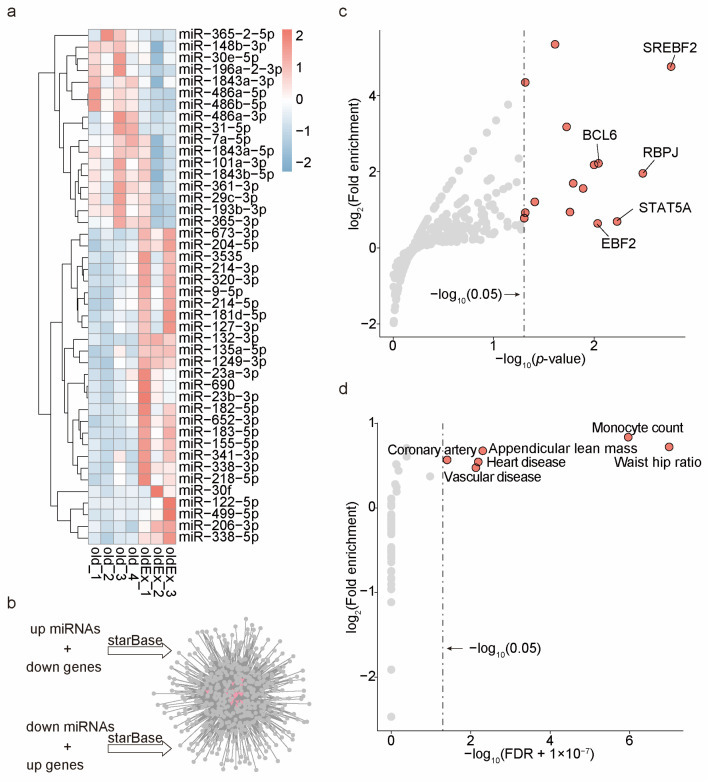
Construction of an exercise-induced miRNA regulatory network and analysis of its upstream TFs. (**a**) Heatmap displaying expression profiles of the DEmiRs (oldEx vs. old, adjusted *p*-value < 0.05) through hierarchical clustering. Each column represents a sample, and each row represents a gene. The color scale indicates the raw Z-score, ranging from red (high expression) to blue (low expression). (**b**) Potential miRNA–target relationships between DEGs and DEmiRs. DEGs and DEmiRs exhibiting inverse expression changes were further filtered by the starBase database based on predictions of targeted degradation. (**c**) The enrichment analysis of upstream TFs for these miRNAs in the network is conducted through a hypergeometric test. Red dots represent significantly enriched TFs (*p*-value < 0.05). The upstream TFs of miRNAs are sourced from the TransmiR database. (**d**) Enrichment analysis of GWAS trait-related SNPs within the TRRs of the network. These SNPs were sourced from the GWAS catalog database and showed significant associations with the target traits, meeting a stringent significance threshold of *p* < 5 × 10^−8^. Red dots indicate FDR < 0.05.

**Figure 3 ijms-25-10773-f003:**
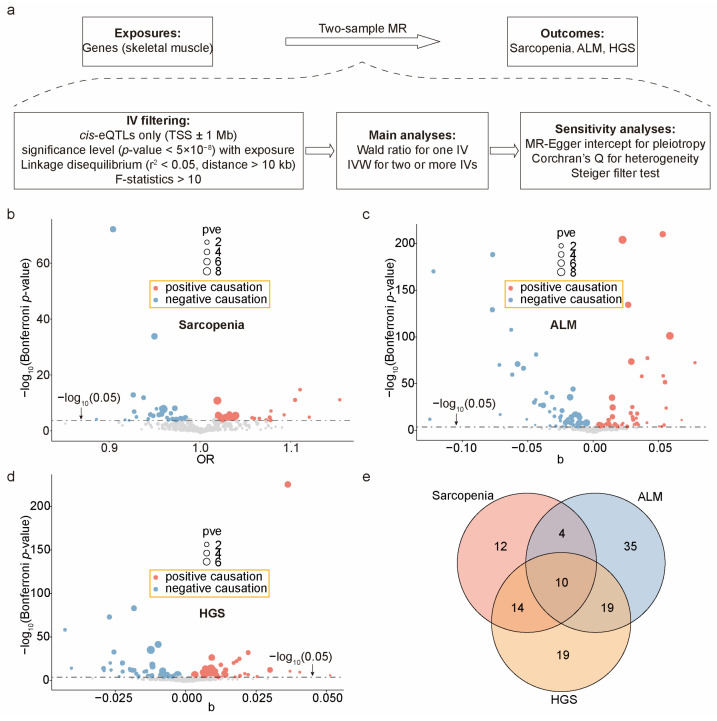
Causal estimates of genes associated with sarcopenia-related traits. (**a**) An overview of the MR framework. The genes under investigation are derived from the miRNA–target network. IV, instrumental variables; *cis*-eQTLs, cis expression quantitative trait loci; IVW, inverse variance weighted. (**b**–**d**) Identification of causal genes for sarcopenia (**b**), ALM (**c**), and HGS (**d**). Genes meeting the significance criteria for MR assessment but exhibiting horizontal pleiotropy or heterogeneity in IVs were excluded. OR (Odds Ratio) is used for binary outcomes, while beta values are used for continuous outcome variables in the MR analysis to quantify the magnitude of causal effects. pve, proportion of variance explained. (**e**) Intersection of causal genes for sarcopenia, ALM, and HGS.

**Figure 4 ijms-25-10773-f004:**
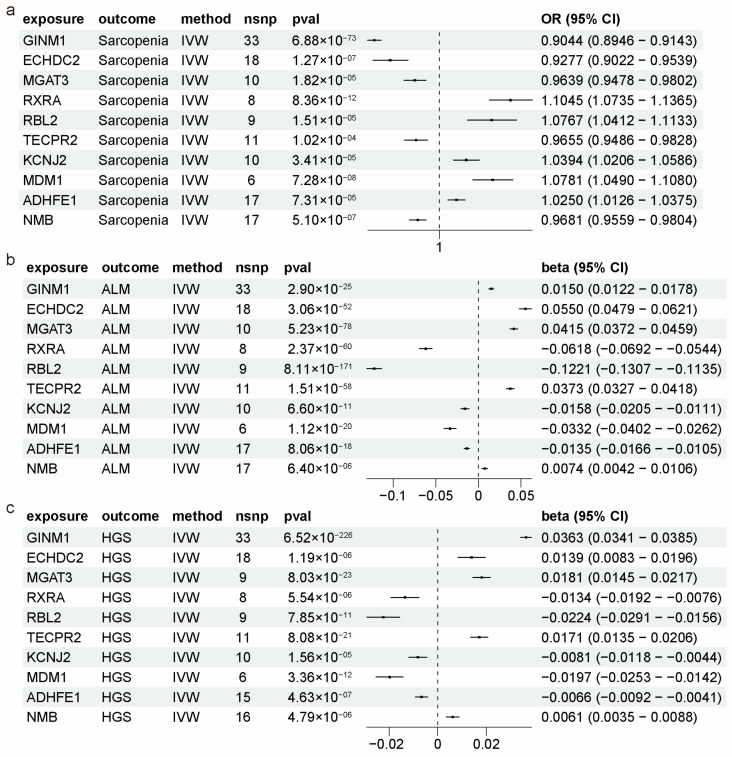
Causal effects of 10 shared genes associated with sarcopenia-related traits. (**a**–**c**) Forest plots showing the two-sample MR estimation of the association between 10 shared causal genes and three sarcopenia-related traits: sarcopenia (**a**), ALM (**b**), and HGS (**c**). nsnp, number of single nucleotide polymorphisms; CI, confidence interval.

**Figure 5 ijms-25-10773-f005:**
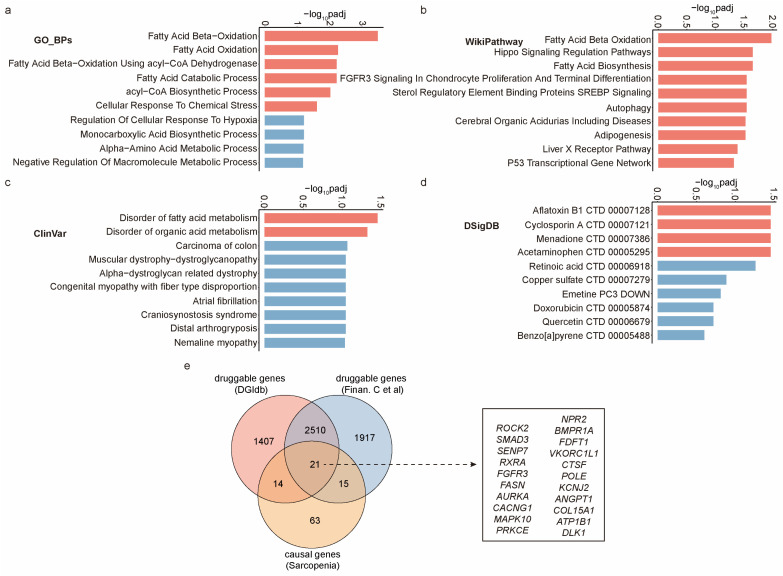
Functions and druggable potential of causal genes in sarcopenia. (**a**–**d**) The Enrichr platform was used to explore the significant biological processes (**a**), cellular signaling pathways (**b**), human diseases (**c**), and drugs (**d**) associated with these genes using the GO Biological Process 2023, WikiPathway 2023 Human, ClinVar 2019, and DSigDB options, respectively. Red indicates a significance level of padj < 0.05, while blue indicates a significance level of padj ≥ 0.05. (**e**) Screening of druggable genes among the 113 causal genes for sarcopenia. A total of 21 druggable genes were identified and are displayed within the rectangle.

## Data Availability

The sequence data presented in this study are openly available in GSA (Genome Sequence Archive) under accession number CRA012450.

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
