# Peer review of "Investigating the Causal Effects of Exercise-Induced Genes on Sarcopenia"

_ijms, 2024, doi:10.3390/ijms251910773_

Round 1

Reviewer 1 Report

Comments and Suggestions for Authors

Dear Authors,

I congratulate you on a, in my opinion, well preformed and well described study. I have one scientific comment and a couple of spelling changes.

- You mention in the section on drugs (line 270->) that most are poor candidates due to side effects. But in line 369-> they are mentioned as potential candidates. Which is it? I think the discussion should be expanded a little here.

- line 93: "moue" should be "mouse" and "gene" "genes"

- line 129: "pearson" is a name so should be "Pearson"

Author Response

We would like to express our sincere gratitude to Reviewer 1 for the constructive and positive comments. Each of these insights has been invaluable in enhancing our manuscript. All authors have carefully discussed the feedback provided. In response to the reviewer’s comments, we have made every effort to modify our manuscript to align with the journal's requirements. In this revised version, the Word document highlights all changes made, and our point-by-point responses to the reviewer are detailed below this letter. The line numbers referenced in all responses correspond to those in the PDF version of the revised manuscript.

Reviewer 1

I congratulate you on a, in my opinion, well preformed and well described study. I have one scientific comment and a couple of spelling changes.

- You mention in the section on drugs (line 270->) that most are poor candidates due to side effects. But in line 369-> they are mentioned as potential candidates. Which is it? I think the discussion should be expanded a little here.

Response:

Thank you for the valuable suggestion. We revisited the section on drug-gene interactions and made corresponding revisions, which significantly improved the manuscript. In the first draft, we identified four drugs interacting with our causal genes using the DSigDB database. However, the drug-gene interactions from DSigDB are based on drug-induced gene expression changes, not on direct binding, and many of these drugs have significant side effects. While this result suggests the potential for druggability among these causal genes, these drugs are not suitable as potential candidates.

To address this, we expanded the druggability analysis to identify druggable genes within our causal gene set. Druggable genes are defined by the presence of drug-binding sites in their protein products. We took the intersection of druggable genes from the DGIdb database and the study by Finan et al., resulting in a robust set of druggable genes. Among our causal genes, 21 were found to be druggable, and DGIdb identified nearly 800 drugs that target these 21 genes. In summary, our study provides 21 potential therapeutic targets and their corresponding drugs, offering a preliminary theoretical foundation for the development of clinical treatment strategies for sarcopenia.

Overall, we have removed the description of Aflatoxin B1, Cyclosporin A, Menadione, and Acetaminophen as 'potential candidates' and added a careful description of the druggability analysis. Details of these revisions can be found in the updated manuscript (PDF version) from lines 252 to 266, and 352 to 358.

- line 93: "moue" should be "mouse" and "gene" "genes"

Response:

The spelling issues have been corrected, as detailed in line 93 of the revised manuscript (PDF version).

- line 129: "pearson" is a name so should be "Pearson"

Response:

The spelling issues have been corrected, as detailed in line 129 of the revised manuscript (PDF version).

Reviewer 2 Report

Comments and Suggestions for Authors

This study aimed to determine effect of aerobic exercise regimen on late middle-aged mice and developed an integrated approach that combines mouse exercise-induced genes with human GWAS datasets to identify causal genes for sarcopenia.  The study also generated results regarding exercise-induced genes and miRNAs. By constructing a miRNA regulatory network enriched with TFs and GWAS signals related to muscle function and traits, the authors focused on 896 exercise-induced genes.

In the GWAS analysis, the authors employed bovine and human genomic databases for analysis of SNP linkage to various traits. However, this might have an issue regarding disconnection different analytic data because of animal species used as the data.

Employing bovine database, the authors obtained the relevant terms with skeletal muscle aging. However, most of these terms are inappropriate terms due to the significance in postmortem beef quality, but not in live skeletal muscle aging. Please see the details described below.

Besides the above major issues, the authors should address minor issues listed below.

line 93: How did you evaluate appropriateness of use of human data for the mouse data analyses? There should be some limitations to be discussed.

line 113: Myh6 and Myh13 are not skeletal muscle specific and should not be treated as the marker.

line 177: The authors employed bovine and human genomic databases for SNP linkage to various traits. However, this might have an issue regarding disconnection different analytic data because of animal species used as the data. I think the authors should address to explain disadvantages due to use of several animal species to explore the phenotypic association in one analysis, as limitation in Discussion.
In addition, mouse was employed as muscle samples in tissue analysis, which might have also disconnection when it would be discussed in association of exercise effect on alteration of gene network. Provide explanation.

line 186: These traits are important traits in meat science  but these are not essential in this study, especially "marbling score" which means results of evaluation of intramuscular fat accumulation in beef. The metabolites in figure 3 a (oleic acid, palmitic acid, myristic acid, myristoleic acid) are also highly accumulated in beef fat, but association of these metabolites with mouse muscle is doubtful and meaningless. How do these meat quality traits relate to mouse muscle characteristics or exercising effect? Also, the authors should consider appropriateness of use of these terms regarding meat quality, including pH. Furthermore, the results of these terms could come from use of inappropriate database to analyze exercising effect on muscle aging. As I mentioned before, at least, use of bovine database is inappropriate, therefore data from bovine database should be removed.

line 190: What is "muscle quality"? How do the authors define it? There is a term "Meat quality" defined speci0fically in meat science and industry, but I have never heard "muscle quality". If this term is not objectively defined or widely accepted, it should not be used. Of course, "Meat quality" is also true,  because no one in civilized countries eat mouse muscle.

line 395: Collection of RNA samples is not explained, which is required.

Author Response

We would like to express our sincere gratitude to Reviewer 2 for the constructive and positive comments. Each of these insights has been invaluable in enhancing our manuscript. All authors have carefully discussed the feedback provided. In response to the reviewer’s comments, we have made every effort to modify our manuscript to align with the journal's requirements. In this revised version, the Word document highlights all changes made, and our point-by-point responses to the reviewer are detailed below this letter. The line numbers referenced in all responses correspond to those in the PDF version of the revised manuscript.

Reviewer 2

This study aimed to determine effect of aerobic exercise regimen on late middle-aged mice and developed an integrated approach that combines mouse exercise-induced genes with human GWAS datasets to identify causal genes for sarcopenia.  The study also generated results regarding exercise-induced genes and miRNAs. By constructing a miRNA regulatory network enriched with TFs and GWAS signals related to muscle function and traits, the authors focused on 896 exercise-induced genes.

In the GWAS analysis, the authors employed bovine and human genomic databases for analysis of SNP linkage to various traits. However, this might have an issue regarding disconnection different analytic data because of animal species used as the data.

Employing bovine database, the authors obtained the relevant terms with skeletal muscle aging. However, most of these terms are inappropriate terms due to the significance in postmortem beef quality, but not in live skeletal muscle aging. Please see the details described below.

Response:

Thank you for your expert suggestions. We have removed the enrichment analysis of bovine SNPs and retained only the human SNPs enrichment analysis. This result has been integrated into Section 2.2, making the manuscript more coherent and the structure more concise. Our initial rationale was that if this miRNA regulatory network is related to skeletal muscle aging, it might also be associated with some basic muscle physiological and biochemical traits. However, studies on such traits in humans are limited by factors like small sample sizes, data heterogeneity, the complexity of data collection, and ethical concerns, leading to a lack of relevant GWAS research. In contrast, livestock, especially cattle, have accumulated a wealth of GWAS data on muscle traits due to long-term breeding and reproductive studies. Therefore, we examined the enrichment of SNPs associated with muscle traits in cattle for orthologous sequences within this network. While some terms showed statistical significance, these terms were known academically as some beef quality indicators, making them unsuitable for association with skeletal muscle aging. As a result, we have removed the analysis of bovine SNPs, which can be found in lines 162 to 176 of the revised manuscript (PDF version).

Besides the above major issues, the authors should address minor issues listed below.

line 93: How did you evaluate appropriateness of use of human data for the mouse data analyses? There should be some limitations to be discussed.

Response:

Mice are the most commonly used model organisms in human disease research and have been extensively employed to explore the mechanisms by which exercise counteracts skeletal muscle aging (Brett JO, et al. Nat Metab. 2020. PMID: 32601609). This is due to the high genetic homology between mice and humans, as well as the similarities in their organs and systems, which perform physiological functions in comparable ways and are susceptible to similar diseases (Xie WQ, et al. J Cachexia Sarcopenia Muscle. 2021. PMID: 33951340). Following a similar approach to previous studies that integrated mouse and human data (Li Z, et al. Cell Metab. 2020. PMID: 32197071), our research combines expression data from mice with human GWAS summary statistics to identify potential causal genes for sarcopenia among exercise-induced genes.

The integration of data from these two species proved successful in this study. For instance, the exercise-induced genes identified in aging mouse skeletal muscle were significantly enriched in human GO terms related to muscle tissue development and fatty acid oxidation (Figure 1d-e). Additionally, the exercise-induced miRNA regulatory network was closely associated with human diseases and traits-related skeletal muscle aging (Figure 2d).

However, there are certain limitations and challenges. Cross-species differences mean that the responses to exercise in mice and humans are not entirely identical. Furthermore, the mouse experiments involved short-term interventions under highly controlled conditions, which do not fully replicate the long-term effects of exercise during the human aging process in more complex environments.

The key contribution of this study lies in its focus on specific targets, identifying 10 potential biomarkers for sarcopenia and 21 potential therapeutic targets. Future research could focus on further investigating the exercise response of these key genes in human skeletal muscle aging, and exploring their potential applications in the diagnosis and prevention of sarcopenia. For more details, please refer to the revised manuscript (PDF version) from lines 359 to 385.

line 113: Myh6 and Myh13 are not skeletal muscle specific and should not be treated as the marker.

Response:

Indeed, Myh6 primarily functions in cardiac muscle, while Myh13 is mainly active in extraocular muscle. We have removed the mention of these two genes in the manuscript, as reflected in the revised manuscript (PDF version) from line 113.

line 177: The authors employed bovine and human genomic databases for SNP linkage to various traits. However, this might have an issue regarding disconnection different analytic data because of animal species used as the data. I think the authors should address to explain disadvantages due to use of several animal species to explore the phenotypic association in one analysis, as limitation in Discussion.

In addition, mouse was employed as muscle samples in tissue analysis, which might have also disconnection when it would be discussed in association of exercise effect on alteration of gene network. Provide explanation.

Response:

In reference to the first and second responses, we fully understand and appreciate the reviewers' concerns regarding the integration of cross-species data. We have made the necessary revisions and added a supplementary discussion. Specifically, we removed the enrichment analysis of cattle GWAS trait-related SNPs. Please see the revised manuscript (PDF version) from lines 162 to 176. This deletion does not affect the main results or conclusions of the paper. We have also addressed the limitations and challenges of integrating mouse and human data, including the limitation mentioned in this comment regarding the use of mouse skeletal muscle samples to identify exercise-induced genes and regulatory networks. Please see the revised manuscript (PDF version) from lines 359 to 385.

line 186: These traits are important traits in meat science  but these are not essential in this study, especially "marbling score" which means results of evaluation of intramuscular fat accumulation in beef. The metabolites in figure 3 a (oleic acid, palmitic acid, myristic acid, myristoleic acid) are also highly accumulated in beef fat, but association of these metabolites with mouse muscle is doubtful and meaningless. How do these meat quality traits relate to mouse muscle characteristics or exercising effect? Also, the authors should consider appropriateness of use of these terms regarding meat quality, including pH. Furthermore, the results of these terms could come from use of inappropriate database to analyze exercising effect on muscle aging. As I mentioned before, at least, use of bovine database is inappropriate, therefore data from bovine database should be removed.

Response:

We have removed the analysis of cattle data, as reflected in the revised manuscript (PDF version) from lines 162 to 176.

line 190: What is "muscle quality"? How do the authors define it? There is a term "Meat quality" defined speci0fically in meat science and industry, but I have never heard "muscle quality". If this term is not objectively defined or widely accepted, it should not be used. Of course, "Meat quality" is also true,  because no one in civilized countries eat mouse muscle.

Response:

The mention of "muscle quality" in the first version of the manuscript pertained to the analysis of cattle data. This section has been removed.

line 395: Collection of RNA samples is not explained, which is required.

Response:

In the original manuscript, line 395 referred to section 4.2, "Histological Investigation," which does not involve RNA sample collection. I believe the reviewer was referring to section 4.3, "Sequencing Library Construction," where the description of RNA sample collection was lacking. This has now been addressed and supplemented in the revised manuscript (PDF version) from lines 410 to 412.

Round 2

Reviewer 2 Report

Comments and Suggestions for Authors

In the present version of manuscript, authors have appropriately addressed to issues that were pointed out. With this version, I recommend acceptance of this article for publication.